# The Role of Hyaluronic Acid in Intervertebral Disc Regeneration

**Zepur Kazezian** [1,†]**, Kieran Joyce** [1,2] **and Abhay Pandit** [1,*]

1    CÚRAM, SFI Research Centre for Medical Devices, National University of Ireland Galway,
     H91 W2TY Galway, Ireland; z.kazezian@imperial.ac.uk (Z.K.); k.joyce10@nuigalway.ie (K.J.)
2    School of Medicine, National University of Ireland Galway, H91 TK33 Galway, Ireland
*    Correspondence: abhay.pandit@nuigalway.ie
†    Zepur Kazezian is currently at Imperial College London, London SW7 2AZ, UK.

**Abstract:** Intervertebral disc (IVD) degeneration is a leading cause of low back pain worldwide, incurring a significant burden on the healthcare system and society. IVD degeneration is characterized by an abnormal cell-mediated response leading to the stimulation of different catabolic biomarkers and activation of signalling pathways. In the last few decades, hyaluronic acid (HA), which has been broadly used in tissue-engineering, has popularised due to its anti-inflammatory, analgesic and extracellular matrix enhancing properties. Hence, there is expressed interest in treating the IVD using different HA compositions. An ideal HA-based biomaterial needs to be compatible and supportive of the disc microenvironment in general and inhibit inflammation and downstream cascades leading to the innervation, vascularisation and pain sensation in particular. High molecular weight hyaluronic acid (HMW HA) and HA-based biomaterials used as therapeutic delivery platforms have been trialled in preclinical models and clinical trials. In this paper, we reviewed a series of studies focused on assessing the effect of different compositions of HA as a therapeutic, targeting IVD degeneration. Overall, tremendous advances have been made towards an optimal form of a HA biomaterial to target specific biomarkers associated with IVD degeneration, but further optimization is necessary to address regeneration.

**Keywords:** hyaluronic acid; disc repair; annulus fibrosus repair; biomaterials; inflammatory biomarkers and signalling pathways

---

## 1. Introduction

A total of 80% of the world population suffer from low back pain (LBP) which is considered the most significant cause of disability, resulting in a negative socioeconomic impact [1,2]. In terms of disability-adjusted life years, LBP incurs a substantial burden over other health-related conditions [3,4]. Although LBP is more common in patients above 65 years old [5,6], it can start as early as in the late teenage years [7]. It is classified as the most expensive healthcare treatment, ranging between 12 to 90 billion dollars in the United States alone [8] and above 500 million pounds in the United Kingdom according to the nice guideline [9].

LBP is frequently associated with the deterioration of the intervertebral disc (IVD) due to abnormal cell-mediated response leading to the stimulation of different catabolic enzymes [10,11] and signalling pathways [12], imbalance in extracellular matrix composition and overexpression of the extracellular matrix-degrading enzymes [13]. The changes in the biomechanical elements, which are represented by unbalanced mechanical loading [14], genetic background [15] and reduced cellular activity, lead to continuous structural failure (Figure 1) [16]. In terms of biochemical changes, the deficiency in nutrient diffusion into the disc due to the calcified cartilaginous endplate (CEP) is a fundamental cause of IVD

degeneration [17]. Additionally, increased elastin concentrations in the annulus fibrosus (AF) [18], and augmented proteoglycan breakdown in the nucleus pulposus (NP) cause a dramatic decrease in disc height and ultimately lead to biomechanical instability [19].

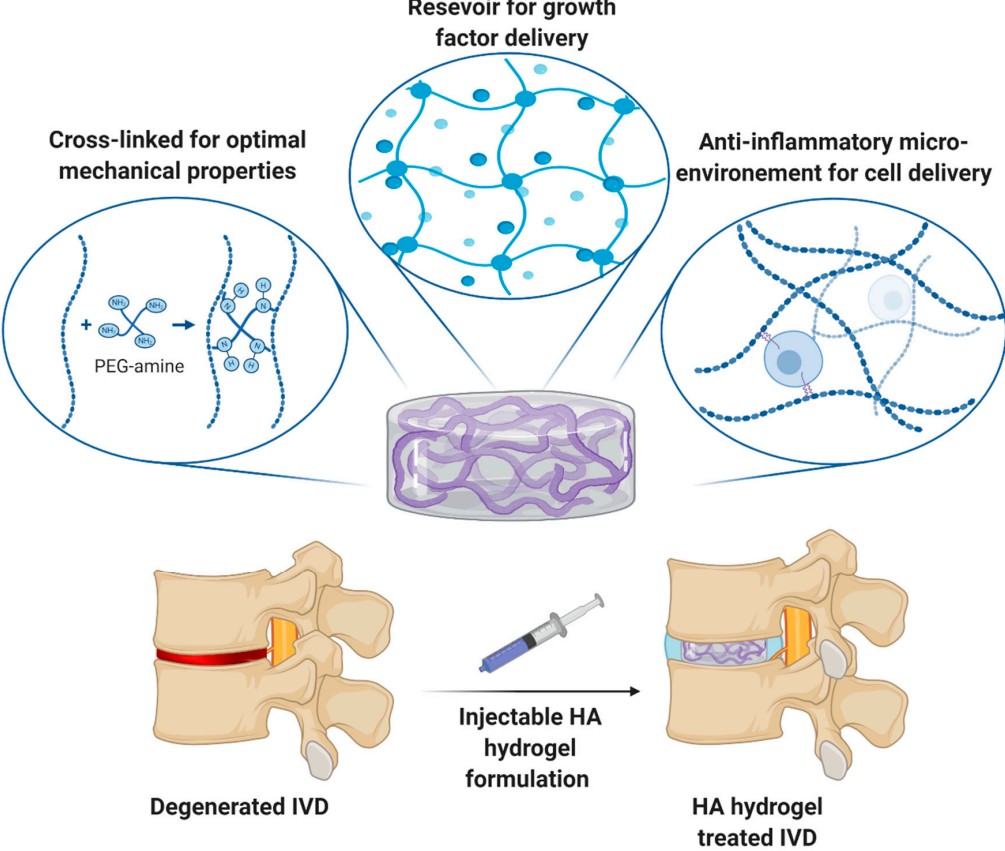

**Figure 1.** Different hyaluronic acid-based hydrogel platforms to treat the intervertebral disc degeneration. Hydrogel properties can be optimized to recapitulate the mechanical properties of the intervertebral disc (IVD), deliver therapeutics including growth factors and cells into the IVD. Hydrogels may be optimized to be injectable for minimally invasive treatment of the IVD.

Treatment of LBP in the early and moderate stages is mostly monitored through conservative treatments including bed rest, physiotherapy and exercise [20,21] which is followed by prescribing non-steroidal anti-inflammatory drugs (NSAID)s including Ibuprofen and COX-2 inhibitors (Celecoxib), that are effective in treating acute and chronic LBP comparing to placebo [22]. In addition to NSAIDs, muscle relaxants such as cyclobenzaprine are also used [23]. Moreover, opioids and benzodiazepines are prescribed to treat LBP. However, they have a short-term effect and need to be prescribed for no longer than four weeks because of the risk of developing dependency [24,25]. Furthermore, epidural and systemic corticosteroids are recommended with clinical trials exhibiting heterogeneous results, with the latest outcomes indicating that glucocorticoid receptor-specific steroids may be more effective comparing to mineralocorticoid-targeting steroids [26].

Although most of the LBP patients will recover after an acute phase, 10% may develop chronic low back pain [27]. In chronic stages, generally, surgery is required because the IVD is unable to cure itself [28]. Existing surgical interventions include discectomy and spinal fusion, or—occasionally in the cervical spine—a total disc replacement is indicated [29]. Patients who had discectomy show improvement when compared to nonsurgical therapy in short-term follow-ups; however, long term follow-ups (beyond two years) indicate no significant difference in the outcomes [30]. Spinal fusion is the typical surgical approach for chronic LBP. Although nonsurgical treatment for chronic LBP may be useful in a particular cohort, lumbar fusion remains more effective in alleviating pain and decreasing

disability than commonly used nonsurgical treatments [31]. Disc replacement is also used to tackle the incidence of the disease in the adjacent segments due to the acute variation in motion segment mechanics linked with anterior cervical discectomy and fusion [32]. There is minimal evidence that suggests early surgical intervention improves long-term results in patients with lumbar disc herniation and radiculopathy that did not progress yet into neurologic deficit [30,33].

These treatments are not deemed curative, nor do they reverse the underlying pathology but rather aim to achieve symptomatic relief. Therefore, tissue-engineering approaches through developing biomaterial-based platforms to deliver active therapeutic products such as mesenchymal cells, inhibitory molecules and growth factors (GF) are being investigated to repair the degenerated IVD and revert it to a healthy state [34]. In the last few decades, hyaluronic acid (HA) was broadly used in treating osteoarthritis, popularised due to its anti-inflammatory, analgesic and extracellular matrix enhancing properties [35–37]. Hence, researchers shifted their interest towards treating the intervertebral disc using different HA compositions to deliver cells and therapeutics in vitro and in vivo (Figures 1 and 2. Therefore, the objective of this review article is to summarise the different HA-based platforms used to date in treating IVD degeneration and LBP. We also summarised the biomarkers and signalling pathways involved in disc degeneration. Moreover, we recapitulate the most popular preclinical models utilized so far in the field of IVD degeneration, emphasizing the fundamental need for more representative models to replicate human pathophysiology.

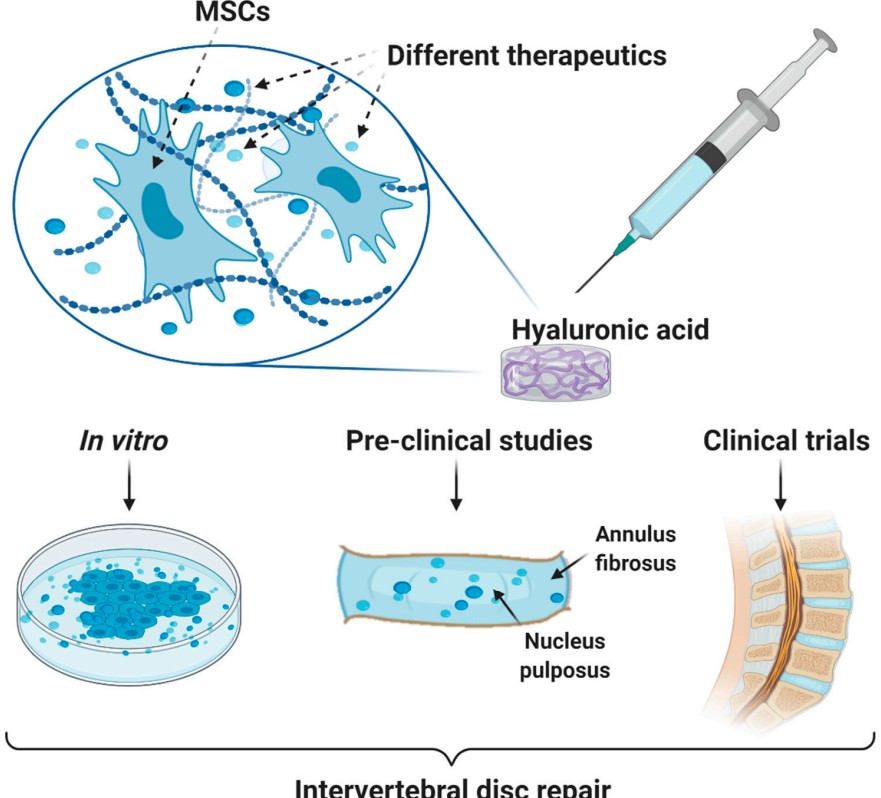

**Figure 2.** The role of hyaluronic acid as a platform for cell and drug delivery in the IVD. HA acts as a vehicle for adipose tissue and bone marrow isolated mesenchymal stem cells (MSC)s and drug delivery for disc repair. This schematic summarises the role of HA as a treatment for low back pain (LBP) and IVD degeneration in cell culture, preclinical studies and clinical trials. HA has been successful in disc repair through its anti-inflammatory, anti-apoptotic, analgesic and matrix modulatory characteristics.

## 2. Biomarkers and Signalling Pathways Associated with the Degenerated Intervertebral Disc

The literature associates IVD degeneration with dysregulated quantities of inflammatory cytokines produced by infiltrating macrophages, neutrophils, T cells, as well as disc cells [38]. During disc degeneration, the infiltration of mast cells and macrophages is enhanced by the vascularisation of the AF, which leads to amplified propagation of inflammatory signals, consequently, leading to LBP [39]. The inflammatory molecules secreted in the IVD, specifically prostaglandin E2 (PGE2), interferon-gamma (IFN-γ), tumour necrosis factor-alpha (TNF-α), and interleukin one alpha/beta (IL-1α/β), interleukin (IL)-17, IL-10, IL-6, IL-8, IL-2, IL-4, [38], stimulate the breakdown of the extracellular matrix (ECM), cell senescence, autophagy and apoptosis of the disc cells [38,40,41]. Pain sensation was also correlated with up-regulated nitric oxide and nitric oxide synthase [39]. Furthermore, the induced pro-inflammatory cytokines were found to stimulate the expression of neurogenic molecules, such as nerve growth factor (NGF) and brain-derived neurotrophic factor (BDNF), which may support the ingrowth of nerves as well as enhance nociception in dorsal root ganglia (DRG)s [38,41]. Hence, chronic inflammation prompts permanent structural and biochemical changes in the IVD comprising ECM degradation, vascularisation and innervation that ultimately leads to LBP [42]. Increased chemokine expression, especially C-C motif ligand 2 (CCL2), has been reported in the degenerated IVD [43]. Cultured AF cells from Thompson-graded discs of grade II-III were able to stimulate higher expression of CCL2 upon treatment with IL-1β than TNFα unlike AF cells from grades IV-V which showed a similar response to IL-1β and TNFα treatment in terms of CCL2 expression [44].

Different key dysregulated genes and proteins were discovered in the degenerated human AF such as interferon-stimulated genes (ISGs), which are interferon-induced proteins with tetratricopeptide repeats (IFIT)1, IFIT2, IFIT3 as well as Insulin-like growth factor-binding protein 3 (IGFBP3) [12]. Overexpression of the anti-proliferative IFIT3 protein and the pro-apoptotic IGFBP3 can negatively regulate the cell fate directly or indirectly, to induce growth arrest and apoptosis in AF cells [45–48]. IFIT3 can act as an anti-proliferative molecule through inducing the cyclin-dependent kinase inhibitors, such as cyclin-dependent kinase inhibitor 1A and cyclin-dependent kinase inhibitor 1B which interact with cyclin dependent kinases (CDK)s in the G1 phase and prevent the cell from proceeding into the S phase of the cell cycle [45]. IFNα was also able to induce IGFBP3 through the signal transducer, and activator of transcription 1 (STAT1) via cascades that have Insulin-like growth factor (IGF)-related anti-proliferative properties or IGFBP3 directly or indirectly induced pro-apoptotic consequences [46–48]. IGFBP3 is a vital controller of IGF availability that inhibits IGF activity, which is associated with osteoarthritis [49]. Moreover, IGFBP3 can sensitize cells to apoptosis indirectly through TNFα and IFNγ [50,51]. Several signalling pathways have also been identified in the disc as being aberrantly regulated, which are illustrated in Table 1.

## 3. Hyaluronic Acid in Treating the IVD: Structure, Synthesis and Turnover

HA, which was first isolated in the 1930s, is a distinctive glycosaminoglycan, which, unlike other glycosaminoglycans such as heparan sulphate, chondroitin sulphate, and keratan sulphate, is non-sulphated [52]. It is widely used in treating osteoarthritis [36,53–58] and in tissue-engineering approaches for disc repair [59,60]. HA (Figure 3) is an abundant polymer of the ECM with a molecular weight of approximately $10^6$–$10^7$ Da in vivo [61]. HA molecule comprises of chains of constantly repeating disaccharides, glucuronic acid and *N*-acetylglucosamine, which are symbolized by GlcA-β (1→3) and GlcNAc-β (1→4) respectively (Figure 3) [62]. On the intracellular aspect of the cell membrane, synthesis of HA occurs through the hyaluronan synthases (HAS) family, comprised of three types of synthases, activated via different GFs to produce diverse forms of polymerized HA [63]. HAS1 and HAS2 produce the high molecular weight (HMW) of HA of 2–4 × $10^6$ Da, while the low molecular weight of HA (LMW) of $1 × 10^5$–$1 × 10^6$ Da is produced by HAS3 [63]. These enzymes are regulated by miRNAs [64] and GFs such as TGF-β [65]. Once synthesized, HA is secreted either on the cell surface from the plasma membrane or into the ECM because of its large size [63,66]. The body composition of

HA is 15 g [67]. Daily, 33% of HA is turned over through several roots: (i). via the lymphatic system or (ii). via the circulation through the liver or (iii). lysosomal digestion after HA binds to its receptor and is internalized [68]. Furthermore, HA turnover can take place as a result of pathological conditions such as tissue injury and remodelling as a result of oxidative stress and free radicals, HA internalization and degradation, and clearance of HA from tissues via the lymphatics and vascular system [69]. Regarding the receptors through which HA binds to be cleared by the liver and lymph nodes, there are particular receptors such as the hyaluronan receptor for endocytosis 1 (HARE1), lymphatic vessel endothelial hyaluronan receptor 1 (LYVE1) and CD44. From the previously mentioned three receptors, HARE1 is considered the major role player through the lymph and vasculature [68]. HA is internalized and degraded when it cannot be removed by the vasculature and lymph [69,70]. In this process, first HA binds to one of its receptors such as receptor for HA-mediated motility (RHAMM), layilin or CD44 after which it is degraded by hyaluronidase-2 [65,69,70], then it is digested into fragments through *N*-glucuronidases and *N*-acetylglucosaminidases to completely break down the remaining glycans into monosaccharides [70].

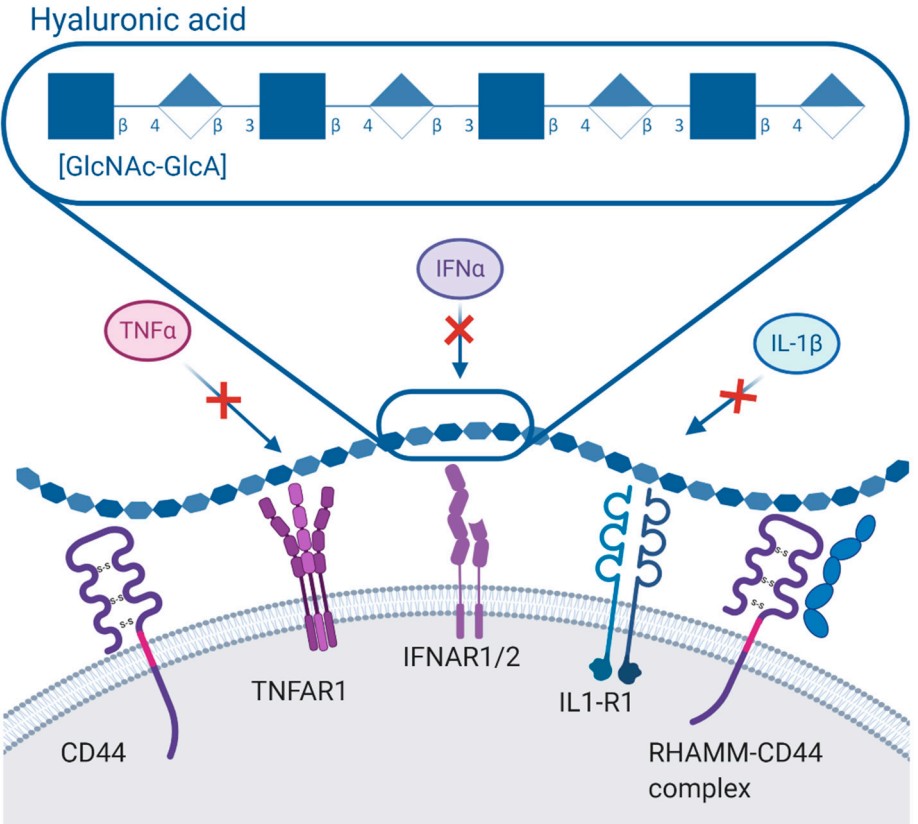

**Figure 3.** The anti-inflammatory role of hyaluronic acid (HA). HA molecule (on the top) is represented according to its international symbols by repeated units of glucuronic acid (GlcA-β1-3) and N-acetylglucosamine (GlcNAc-β1-4) disaccharides. The anti-inflammatory role of HA is mediated through reacting with its several receptors such as CD44 and receptor for HA-mediated motility (RHAMM) preventing the cascades signalled by inflammatory cytokines such as IFNα, TNFα and IL1β through deterring the contact with their receptors.

*The Key Function of Hyaluronic Acid in Repairing the Intervertebral Disc*

The biochemical properties of the HA molecule are entirely dependent on its size [69,71,72]. Within tissues, the HA molecule interacts with various proteins comprising of hyaladherins such as CD44, RHAMM (CD168) (Figure 3). In contrast, inside the ECM, it interacts with N-terminals of proteoglycans such as aggrecan (ACAN), versican, brevican, neurocan and TNFα-stimulated gene-6

(TSG-6) protein [69,73]. These fundamental interactions are essential for HA function in terms of cell-to-cell communication and signalling [65,74]. The mechanical properties of HA can be attributed to its high molecular weight structure. HA is capable of absorbing water approximately $10–10^4$ times its mass as it is negatively charged and is considered an osmotically active molecule [63,64,75,76]. Due to its large volume while hydrated, it can fill large spaces, therefore absorbing shocks and acting like a lubricant [75,77].

HA-CD44 interactions mediate endocytic activity and activate Rho and Rac1 GTPases to regulate cytoskeletal organization [78]. CD44 activation also activates src-related tyrosine kinases to induce cell proliferation through NF-κB signalling [79]. RHAMM activates pp60c-src tyrosine kinase to modulate focal adhesions for RHAMM-mediated cell mobility [80]. Its ability to form large extracellular networks through cell–matrix binding inhibits cytokine binding to prevent downstream signalling (Figure 3) [75,81]. Therefore, HMW HA is capable of inhibiting the signalling of inflammatory cytokines and matrix-degrading enzymes (TNF-$\alpha$, IL-8, iNOS, aggrecanase 2, matrix metalloproteinases (MMP)s) that were evident upon treating chondrocytes with HMW HA, which also resulted in inhibition of phagocytosis and macrophage activation [82,83]. In tissue injury and remodelling, enzymes break down HMW HA resulting in fragments of LMW HA which can induce phosphorylation of p38 or p42 or p44 mitogen-activated protein kinase (MAPK) and NF-κB which are associated with toll-like receptor (TLR)4 signalling [63,84]. Depending on the tissue type and the size of LMW HA fragments (f-HA)s, pro-inflammatory cytokines can be induced [63,81]; therefore, angiogenesis and tissue remodelling is stimulated [69,71]. It was discovered that human disc cells treated with 6–12 disaccharide fragments of HA can dramatically up-regulate catabolic enzymes such as IL-1β, IL-8, IL-6, MMP1, MMP13, and cyclooxygenase (COX2). Furthermore, it was identified that TLR-2 mediated the role of f-HAs in stimulating the protein expression of IL-6, and both IL-6 and MMP-3 were supported by the MAPK signalling pathway [85].

## 4. Translational Preclinical Models Used to Investigate the Intervertebral Disc Degeneration

Studying IVD degeneration using human subjects is challenging because of the ethical restrictions hindering the access of the clinical data as well as differentiating the different factors influencing the disc degeneration which include ageing, genetic predisposition, mechanical loading, and inflammation [86]. Therefore, developing reliable and representative animal models is necessary. As a result, different animal models emerged comprising rodents, canines, porcine, rabbit, sheep, and goat even though these animals are different in their anatomy, development, and tissue mechanical characteristics compared to human discs [87,88]. Such models were utilized to evaluate different hyaluronan-based therapeutic approaches for the IVD repair (Table 2). Several disc degeneration replicas are present to study disc degeneration in in vivo models: spontaneous (genetic aberration induces early disease onset) [89] puncture (physical trauma activating inflammatory response) [90,91] mechanical loading (non-traumatic induction) [92,93] and biochemical (Chondroitinase ABC treatment) [94]. Preclinical models were also used to introduce novel surgical approaches, especially in large animals [95]. Rodent tails have been extensively used to study the pathophysiology of the IVD degeneration and assess the effect of different therapeutics because they are easily accessible with minimal interference with the surrounding tissues and their normal physiological functions [92,93,96–104]. The typical age of the rats used in such studies is three-months-old and above because they can only reach skeletal maturity at that age [105]. Notochordal cells can be found in the IVD of rats throughout adulthood while they are lost in humans [93]; however, these cells are lost in many degenerative rat-tail models due to factors such as mechanical loading [106]. Yurube et al. developed a model of IVD degeneration in a rat-tail in vivo model by using Ilizarov apparatus to load the tail discs, which was first used by Iatridis et al. in 1999 [107]. The Ilizarov-type apparatus was set on the coccygeal 8–9 (C8/9) discs to induce loading on C9/10 while C10/11 remained unloaded as a control. As a result, it was found that after seven day, the notochordal cells were lost in the rat-tail discs upon static compression up to day 56. Moreover, on day seven, it was noted that the extrinsic apoptotic pathway was activated in the AF and NP that

progressively decreased. In contrast, the intrinsic mitochondrial pathway remained continuously active until day 56 [108]. Under disc compression, ECM turnover was explained through tissue inhibitors of metalloproteinase one (TIMP-1), MMP-2 and MMP-13 up-regulation [109,110]. In parallel, it was reported that disc loading leads to imbalances in the manifestation of MMPs, a disintegrin and metalloproteinase with thrombospondin motifs (ADAMTS), and TIMPs in rat-tail discs [92]. Small and large animal models were utilized over the decades to target dysregulated biomarkers and signalling pathways (Table 1) to mitigate the IVD degeneration using HA-based biomaterials summarised in Table 2. Different HMW HA and HA-based therapeutic platforms were tested in vitro, in vivo and human trials to repair the disc (Figure 2).

**Table 1.** Dysregulated signalling cascades in the degenerated IVD.

| Signalling Cascades | IVD Segment | Reference |
|---|---|---|
| Transforming growth factor-beta (TGF-β) | IVD | [111] |
| Mammalian target of rapamycin (mTOR) | IVD | [112] |
| Wingless-related integration site (Wnt) | AF, NP, end Plate (EP), growth plate (GP) | [113–118] |
| Nuclear factor-kappa beta (NF-κB) | AF, NP, IVD | [119–125] |
| Nerve growth factor (NGF) | AF, NP | [126–131] |
| Mitogen-activated protein kinase (MAPK) | AF, NP | [132–138] |
| Notch signalling (NOTCH) | AF, NP | [139] |
| Fibroblast growth factor (FGF) | AF, NP | [140,141] |
| Interferon-alpha signalling canonical pathway (IFNα) | AF | [12,60] |

**Table 2.** Different hyaluronic acid compositions evaluated in preclinical and clinical trials for IVD repair.

| HA Composition and Cross-Linking Method | Species/Platform | Objectives | Outcome | Reference |
|---|---|---|---|---|
| **Collagen-I and HA** Ultraviolet (UV) cross-linking | Bovine In vitro | To assess the efficacy of collagen-I and HA injection in IVD repair, both including and excluding GF. | • In terms of ECM components, there were no evident changes between AF and NP in synthesizing collagen, aggrecan and leucine-rich proteoglycans. <br> • In both types of cells, TGF-$\beta$1 induced ECM synthesis. <br> • According to the outcomes, the effect of TGF-$\beta$1 and primary fibroblast growth factor (bFGF) combination was more effective than that of TGF-$\beta$1. | [142] |
| **HA and chondroitin sulphate (CS) hydrogel** Cinnamic acid + UV cross-linking | Rabbit In vivo | To evaluate the cross-linked HA and CS's effect on the disc regeneration. | • Histological staining showed that the inner AF of the discs that were treated with cross-linked HA or CS have thick Safranin-O staining which is than those injected with 1% sodium hyaluronate or phosphate buffer saline (PBS). <br> • According to the magnetic resonance imaging (MRI) results, the injection of cross-linked HA and CS can neutralize the intensities of MRI signals detected by injured discs. | [143] |
| **Hylan G-F20 (Synvisc)** Avian derived HA Cross-linked by divinyl sulfone | Primate In vivo | To evaluate if HA can affect the degenerative pathway in non-human primates after undergoing nucleotomy. | • Comparing to the control discs, Hylan G-F20 administered segment height was reduced of around 10% only unlike the ones undergone nucleotomy. <br> • Comparing to the control, standard nucleotomy and Hylan B treated functional spinal unit resulted in significant disc space narrowing. <br> • According to the MRIs, CT scans, and macroscopic evaluation, Hylan G-F20 administration suggested to be significantly effective in comparison to the standard nucleotomy when compared to the control. <br> • Overall, Hylan G-F20 can be a postoperative therapeutic approach to halt disc degeneration after nucleotomy. | [144] |
| **HMW HA** PEG-amine cross-linked with (1-ethyl-3-(3-dimethylaminopropyl) carbodiimide hydrochloride) (EDC)/N-hydroxysuccinimide (NHS) | Bovine Ex vivo | To identify HA's matrix modulatory and anti-inflammatory properties. | • The outcomes revealed that HMW HA demonstrated anti-inflammatory properties by decreasing the expression of IGFBP3 and IFIT3 as well as the downstream signalling molecules of IFN$\alpha_2\beta$. <br> • HMW HA was also able to up-regulate the ECM aggrecan and collagen-1. | [145] |

**Table 2.** *Cont.*

| HA Composition and Cross-Linking Method | Species/Platform | Objectives | Outcome | Reference |
|---|---|---|---|---|
| **HMW HA** **PEG-amine cross-linked with EDC/NHS** | Bovine In vitro | To identify the effect of HA in an IL-1β inflammation model of NP cells. | <ul><li>HA hydrogel was stable in PBS and retained more than 40% mass under degradation by enzymes.</li><li>There was no observed cytotoxic effect of HA hydrogel on NP cells for the duration of the culture up to 7 days.</li><li>Gene expression analysis revealed that upon treating cells with cross-linked HA, there was a significant down-regulation of IL-1R1, myeloid differentiation factor 88 (MyD88), NGF and BDNF.</li><li>Upon treatment with HA, CD44 receptors were up-regulated in the NP cells after which suggested that HA can have an anti-inflammatory effect by reacting with CD44 on the cells.</li></ul> | [59] |
| **Glycidyl methacrylate (GM)-HA** **HA with collagen devoid matrix** **UV cross-linking** | Porcine In vivo | To evaluate HA/collagen hydrogel in a porcine nucleotomy model. | <ul><li>IVD degeneration was induced after nucleotomy using a 16G needle.</li><li>An inflammatory reaction to the material was detected.</li><li>Inflammation leads to annular scarring. However, the hydrogel was successful in deterring re-herniation.</li><li>HA-treated discs increased expression of collagen-I collagen-II, MMP13 and TIMP1.</li></ul> | [146] |
| **Hyaluronan gel (Durolane®)** | Porcine In vivo | To evaluate HA as a vehicle for different types of cells used for IVD repair. | <ul><li>The hyaluronan gel promoted cell proliferation in vitro.</li><li>Synthesis of collagen-II was observed in the MSCs and chondrocytes which survived in the porcine IVDs up to six months.</li><li>MRI scanning revealed significant changes in the endplate showing severe IVD degeneration in animals after six months of transplanting various types of implants.</li><li>Bone mineralization was evident by positive staining.</li><li>In vivo tested hyaluronan gel was not successful as a cell carrier.</li></ul> | [147] |
| **HMW HA** **PEG-amine cross-linked with EDC/NHS** | Rat In vivo | To assess HA's effect on the IFNα expression as well as the extracellular matrix modulation by conducting proteomic data analysis. | <ul><li>IFNα was significantly down-regulated on days 7, 28 and 56 in the discs bearing injury, which were implanted with HMW HA.</li><li>Caspase 3 was found down-regulated on days 7, 28 and 56 in the discs bearing injury which were implanted with HMW HA.</li><li>In ECM, two key ECM proteins were found up-regulated, which are aggrecan and hyaluronan and proteoglycan link protein 1 (HAPLN1) over the different time points in response to HMW HA treatment.</li><li>Regarding the glycosylation pattern, we found sialylation, which is an indicator of inflammation was down-regulated on days 7, 28 and 56 when the injured discs were treated with HMW HA.</li></ul> | [60] |

**Table 2.** *Cont.*

| HA Composition and Cross-Linking Method | Species/Platform | Objectives | Outcome | Reference |
|---|---|---|---|---|
| **HMW HA** <br> **PEG-amine cross-linked with EDC/NHS** | Rat <br> In vivo | To assess HA's effect as an analgesic and anti-inflammatory therapeutic in a pain model of IVD. | • HA reduced nociceptive behaviour and inhibited, hyper-innervation. <br> • HA altered the glycosylation pattern of the degenerated disc. <br> • HA modulated key inflammatory signalling pathways; therefore, attenuating inflammation and its effect on the extracellular matrix. | [148] |
| **Aminated hyaluronic acid-g-poly N-isopropyl acrylamide-(AHA-g-PNIPAAm)-gefitinib** <br> **EDC cross-linking** | Mice/Rat/ human <br> In vivo | To assess the effect of AHA-g-PNIPAAm-gefitinib in mitigating IVD degeneration by intervening with the function of the epidermal growth factor receptor (EGFR). | • AHA-g-PNIPAAm-gefitinib suppressed EGFR activity, increased autophagy and ECM production, decreased MMP13 and prevented the progression of IVD degeneration in humans. | [149] |
| **Hyaluronic and fibrin acid hydrogel (RegenoGel)–Fibroblast growth factor-18** <br> **Fibrin cross-linking** | Human/ Bovine <br> In vitro | To assess the effect of both FGF-18 and the FBG-HA hydrogel on NP regeneration. | • FBG-HA-treated human NP cell cultures showed an up-regulation in the expression of carbonic anhydrase XII and collagen-II after 7 and 14 days, respectively. <br> • Increased glycosaminoglycan (GAG) release was noted over 14 days in the conditioned medium. <br> • From day 7 to day 14, increased expression of ACAN was noted in bovine NP cells. <br> • In human NP cells, FGF-18 up-regulated CA12. <br> • According to the histology results: 1. Unlike human NP, bovine cells showed an increase in proteoglycan synthesis as a result of FGF-18. 2. FBG-HA hydrogel showed a significant effect on mitigating the degeneration. <br> • In terms of cell proliferation and GAG synthesis, FGF-18 did not show any significant effect in the NP cells. | [150] |
| **HYADD®4-G** <br> **hexadecyl amine of $5 \times 10^2$ to $7.3 \times 10$ kDa HA aliphatic amines bound to glucuronic acid at 2% substitution** | Rabbit <br> In vivo | To evaluate the effect of HYADD®4-G in IVD repair. | • Compared to the control, saline injections increased the disc height by 50% of the initial disc height while HYADD®4-G administration increased it by over 75%. <br> • MRIs showed that HYADD®4-G administration leads to significantly higher water absorption compared to the control treatment. <br> • HYADD®4-G injected discs (83.3%) were restored to grade I based on the MRI grading system (Pfirrmann). <br> • In terms of cellularity, tissue organization and comparing to saline injection, HYADD®4-G administration showed significantly decreased scores of IVD degeneration. <br> • No inflammatory reactions were observed to HYADD®4-G injection into the discs. | [151] |

**Table 2.** *Cont.*

| HA Composition and Cross-Linking Method | Species/Platform | Objectives | Outcome | Reference |
|---|---|---|---|---|
| **Platelet-rich plasma (PRP) and HA Batroxobin (BTX) cross-linking** | Human In vitro | To assess whether PRP/HA/BTX blend is the best delivery vehicle for MSCs. | • There was higher MSCs viability and proliferation in the hydrogel.<br>• Compared to the control, significantly higher GAG production was observed in MSCs in culture, including or excluding TGF-β1.<br>• Histology and gene expression analysis revealed that MSCs within the hydrogel treated with TGF-β1, differentiated into chondrocyte-like cells expressing ACAN, COL2 and SOX9. | [152] |
| **Hyaluronan oligosaccharides (HA-oligos)** | Ovine In vitro In vivo | To evaluate the effect of HA-oligos in inducing ECM anabolic gene expression and metalloproteinase. | • AF cells cultured in 2D respond mildly to the HA-oligo, where proMMP-2 levels were slightly up-regulated while MMP-9 was not changed; in NP cells, ProMMP-2 increased in a dose-dependent manner.<br>• In AF alginate bead culture, the active form of MMP-9 and pro-MMP-2 was up-regulated until day 10; while in NP alginate bead culture, active MMP-9 was up-regulated on day ten while proMMP-2 continuously changed into the active MMP-2 in days 7–10.<br>• In the non-stimulated NP cultures, MMP-2 and MMP-9 activity were down-regulated over 2–10 days.<br>• HA-oligo was shown not to be cytotoxic because of the high disc cell viability.<br>• RT-PCR showed that MMP1, MMP13 and ADAMTS1 as well as the matrix genes COL1A1 and COL2A1 and ACAN were up-regulated in the NP via HA-oligos; while in the AF ADAMTS1, ADAMTS4, ADAMTS5, MMP13, and COL2A1 and ACAN expression were decreased.<br>• Histological analysis showed that in the outer lesion zone, collagen-I enhanced remodelling by the HA-oligo treatment. | [153] |
| **Fibrin/HA (FB/HA) Transglutaminase cross-linking** | Goat In vivo | To evaluate the safety and the effect of injecting bone morphogenic protein 2 (BMP-2) and BMP-2/7 incorporated into (FB/HA) intradiscally for the purpose of disc repair in a mild disc degeneration model in the goat. | • No adverse effects or heterotopic bone formation were observed.<br>• After induction of mild disc degeneration, a significant disc height decrease was observed in the radiographs.<br>• MRI T2* results were correlated with histology and biochemistry outcomes.<br>• Intervention groups did not show any significant changes.<br>• Although BMP-2 and BMP-2/7 turned to be safe, there was no evidence of disc regeneration. | [154] |

**Table 2.** *Cont.*

| HA Composition and Cross-Linking Method | Species/Platform | Objectives | Outcome | Reference |
|---|---|---|---|---|
| **Gelatin and HA methacrylate (GelHA) UV cross-linking** | Rat In vitro In vivo | To evaluate the effect of GelHA on enhancing the adipose stromal cells (ASCs) differentiation. | • GelHA hydrogel induced ASCs differentiation which was detected through up-regulation of the NP markers in the group of GelHA and ASCs compared with the control and ASC alone.<br>• Compared with normal cultured cells, the group including GelHA and ASCs up-regulated TGF-β1 and TGF-β RII genes in 14 and 21-days.<br>• neutralizing antibody suppressed the expression of NP matrix proteins in ASCs in vitro.<br>• In vivo trials in rats showed that the hydrogel composed of GelHA and ASCs enhanced IVD repair through up-regulating NP matrix synthesis and significantly increasing the disc height. | [155] |
| **Collagen and HA incorporated fibrin-based hydrogels Fibrin-thrombin cross-linking** | Porcine In vitro | To evaluate the effect of the HA and collagen incorporated fibrin hydrogel in NP-like matrix synthesis. | • High fibrin concentrations enhanced cell viability and proliferation.<br>• In Fibrin–collagen hydrogels collagen synthesis was also detected.<br>• HA enhanced the chondrocyte proliferation and induced the proteoglycan synthesis in the ECM.<br>• Incorporation of HA enhanced GAG synthesis while led to the suppression of total collagen development at higher concentrations.<br>• 5 mg/mL HA was the most optimal concentration of HA for disc regeneration, and this is because it could support NP-like matrix synthesis of the articular chondrocytes. | [156] |
| **Combined adipose-derived mesenchymal stem cells (AT-MSCs) and HA derivative Butanediol diglycidyl ether cross-linking** | Human Clinical Trial | To assess the effect and tolerability of administration of AT-MSCs combined with HA derivative intradiscally in chronic discogenic LBP patients. | • The clinical trial did not result in any procedural or stem cell-related adverse changes during the follow-ups of the 1st year.<br>• In terms of the Short Form 36 (SF-36), visual analogue scale (VAS), and the Oswestry disability index (ODI) scoring, no significant changes were observed between the low and high cell dose groups.<br>• Three patients out of the six with the cases 4, 8, and 9 who had significant improvement which was prevalent through SF-36, VAS, ODI and scoring, were identified having higher water content in MRI scans.<br>• Results showed that it is safe to implant HA with AT-MSCs in patients with chronic LBP. | [157] |
| **Sodium hyaluronate (SH)** | Human Clinical Trial | To assess the effect and the tolerability of SH administration in comparison with glucocorticoids triamcinolone acetonide (TA) in treating non-radicular back pain. | • After the administration of SH and throughout the follow-up period, there were no unfavourable changes.<br>• The SH treatment, the effect of which was equal to a course of TA injections, improved significantly patients' quality of life of with non-radicular pain via reducing pain and improving function.<br>• Comparing to TA-treated, SH-treated patients expressed more extended-term benefits. | [158] |

There is an increasing requirement for three-dimensional (3D) scaffolds to regenerate IVDs. HA has a diverse range of applications when integrated into such 3D structures. Different materials were incorporated with HA to form composites with biocompatible properties including biological performance, stiffness, and degradation both in vitro and in vivo. Such composites, including collagen, Fibrin and chondroitin sulphate [142,143,146,156], have been investigated in order to tackle extracellular matrix degradation by enhancing the synthesis of key matrix modulating proteins, GAGs and increasing cellular viability. Pre-clinical models of IVD degeneration to test the efficacy of HA-based materials have been developed in mice [155], rats [60], rabbit [143], sheep [153], goats [154], pigs [156], cows [142], and primates [144]. HA formulations have been cross-linked using many methods including UV cross-linking [142], divinyl sulfone [144], polyethyleneglycol (PEG)-amine with EDC (1-ethyl-3-(3-dimethylaminopropyl)carbodiimide), NHS (N-hydroxysuccinimide) cross-linking [59], fibrin [150], batroxobin [152], butanediol diglycidyl ether [157] and enzymatic cross-linking using transglutaminase in a fibrin/HA material [154]. HA hydrogels utilized in treating the different parts of the IVD, including AF and NP, not only acted as composite materials but also as a reservoir for growth factors and therapeutic drugs [142,150,154], as well as a cell delivery vehicle [147,152,155].

In summary, these studies report on the hydrophilic properties of HA to restore disc height [151] and MRI signal intensity [144], the ability of HA to promote ECM synthesis [147] in resident or injected cells, and of course the anti-inflammatory [145] and anti-nociceptive [148] properties of HA in the setting of IVD degeneration. Several clinical trials investigating the efficacy of HA-based materials are ongoing, with few reporting results at this early stage. However, initial results have been promising, where MSCs delivered in a HA carrier system demonstrated an excellent safety profile with several patients reporting improved VAS and ODI reflecting better MRI signal intensity [157].

Furthermore, HA is being trialled in extra-discal injections in cases of facet joint degeneration [158]. HA has demonstrated equivalent efficacy to glucocorticoid administration in this patient cohort. Many studies are yet to report results, including a bone marrow-derived MSC (BM-MSC)/HA system in a Phase 2 trial (NCT01290367), Rexlemestrocel-L combined with hyaluronic acid in Phase 3 (NCT02412735) and discogenic cells in a HA delivery system with study arms in USA and Japan in Phase 1 (NCT03347708). There is much excitement surrounding these systems, given the proven HA preclinical efficacy.

## 5. Discussion and Future Directions

Discogenic back pain is a prevalent disease described by the degeneration of one or more of the intervertebral discs causing pain, dramatically affecting the quality of life. LBP generation is initiated by low numbers of native IVD cells, which leads to extracellular matrix disintegration, and vascularisation promoting inflammation and low back pain sensation. It was found that pro-inflammatory cytokines such as IFN$\alpha$ induce cellular arrest and apoptosis of the disc cells through downstream targets such as IFIT3 and IGFBP3; therefore, leading to low cellularity of the disc [12]. Moreover, the up-regulation of different inflammatory cytokines can induce the disintegration of the ECM, synthesis of chemokine and, ultimately, changes in IVD cells [38].

For the ideal HA-based therapeutic to be identified, it needs to be compatible and supportive to the disc microenvironment in general and target inflammation and downstream signalling cascades that lead to innervation, vascularisation and pain sensation in particular. In the last few decades, HA and a combination of HA-based biomaterials were tested in preclinical disc degeneration models and human clinical trials. Multiple cross-linking techniques have been employed, including UV cross-linking [142], chemical cross-linking [145] and combination devices that also utilize molecular self-assembly [146]. There is no apparent consensus on optimal cross-linking methods, nor has there been detailed investigation into the retained bioactivity using these cross-linking methods. Although it could be challenging because of the complexity of the disease, research has focused on a different combination of materials that could be quite promising for IVD degeneration. HA, a sizeable hydrophilic molecule that can absorb shocks, is a fundamental component of ECM which developed to be a promising platform for next-generation therapeutics. Through the use of various combinations of

HMW HA and processing techniques, inflammation, innervation, as well as pain, can be reduced in the disc [59,60,148]. The material design process must further consider the degradation products of these materials and ensure these metabolites do not induce pro-inflammatory signalling in long-term follow-up. While LMW HA fragments induce inflammation, HA can be modified to resist degradation and mitigate the effects of fragmentation [84,159].

In this review, we identified a series of studies focused on assessing the effect of different compositions of HA as a therapy for aberrantly expressed enzymes, proteins or signalling pathways identified in the degenerated IVD. It was found that HA was effective in lowering the IFN$\alpha$ inflammatory signalling pathway and downstream key apoptosis and cell senescence regulators such as IFIT3 and IGFBP3 in the AF [60]. Similarly, HA was found to be capable of lowering the NF-$\kappa$B signalling pathway and its downstream targets NGF and BDNF in the NP [59].

In addition to its use as anti-inflammatory, anti-apoptotic and analgesic biomaterial in the disc, HA was also used as a carrier for MSCs in vivo [147,155]. The results of recent clinical trials [157] supported that the implantation of adipose-derived MSCs embedded in HA derivative (Figure 2) is non-toxic and endurable in LBP and could enhance the patients' quality of life by reducing the back pain and enhancing their function [158]. HA encapsulation of MSCs injected into the disc offers structural integrity and allows for cytoskeletal attachment through CD44 binding [59,160]. HA may also block noxious signalling pathways present in the degenerated IVD to preserve cell phenotype. Ongoing clinical trials are using HA as a delivery medium for BM-MSCs and discogenic cells to improve cell survival and reduce inflammation [161].

Furthermore, HA has been used as a vehicle to deliver different types of therapeutics (Figure 2) such as TGF-$\beta$1 [142], BMP-2/7 [154] and gefitinib [149]. The swelling capabilities and tuneable mechanical characteristics of HA hydrogels popularise them for the tissue-engineering approach for direct repair of the disc or as a vehicle for cell therapy of small-molecule drug delivery. HA also acts as a vehicle for GF regulating protein synthesis and cell proliferation [162]. HA and its derivatives effectively bind GF through electrostatic interactions to induce signalling and modulate ligand–receptor activity [162]. The degradation of the HA macromolecule by native tissue hyaluronidases can enhance small-molecule release [163].

In general, the majority of the HA applications were investigated in small or large animal models which are limited in their translation of IVD degeneration in humans [93]. These models were not spontaneous disc degeneration models but induced acute inflammatory models through puncture [164,165] or enzymatic digestion [166]. Therefore, these models might not be ideal for testing therapeutics because they induce rapid degeneration of the disc, rather than a natural onset of disease. Hence, a model of IVD degeneration which replicates the human IVD degeneration pathology is necessary.

Next, what is expected in the prospective studies is the assessment of the effect of potentially identified HA-based therapeutics on a larger scale and in the context of the cross-talk of multiple inflammatory signalling pathways to address the complex environment of the degenerative disc disease and support achieving the ultimate goal which is finding a universal HA-based treatment for intervertebral disc degeneration. HMW HA which was widely used by numerous studies as an anti-inflammatory and matrix enhancing molecule [167,168] must be further investigated in a spontaneous model of disc degeneration in a large animal model to address the anti-inflammatory effect on combined key identified inflammatory signalling pathways such as TNF$\alpha$, IFN$\alpha$ and IL1-$\beta$ as illustrated in Figure 3. Another critical factor to focus on is the development of a robust minimally invasive technique for HA administration without requiring open surgery. Although there are challenging obstacles ahead, HA-based injectable hydrogels are strong candidates as effective treatments for disc degeneration.

## 6. Conclusions

The review summarises the different HA-based biomaterials used to repair the IVD degeneration. Different biomarkers and signalling pathways have been identified in the degenerated IVD. However, still, there is no clear understanding of the cross-talk between these signalling pathways or which are the upstream regulators. To date, the tissue-engineering and biomaterials fields have provided significant efforts in treating disc degeneration. Among these biomaterials, HA has shown potential in targeting inflammation. As a result, several HA-based biomaterials have been utilized to address the dysregulation of different inflammatory cytokines and associated downstream signalling pathways to enhance disc repair and revert to tissue homeostasis. Overall, significant improvements have been made towards an optimal and biocompatible form of HA-based platform to deliver drugs and MSCs to target specific biomarkers associated with the degenerated disc. However, a minimally invasive HA-based therapeutic to target intervertebral disc degeneration remains the goal of the future tissue-engineering approaches.

**Author Contributions:** Z.K. designed, wrote, revised and corrected this review article. K.J. wrote, revised and corrected this review article. A.P. supervised, revised and edited this review article. All authors have read and agreed to the published version of the manuscript.

**Funding:** This research was funded by Hardiman Scholarship, NUI Galway, AO Foundation and Science Foundation Ireland for supporting Dr Zepur Kazezian and College of Medicine Nursing and Health Sciences Scholarship, NUI Galway and the European Regional Development Fund and Science Foundation Ireland under Ireland's European Structural and Investment Fund, Grant Number 13/RC/2073, for supporting the Ph.D. studies of Kieran Joyce. All figures were created using biorender.com.

**Acknowledgments:** The authors would like to acknowledge the funding bodies: Hardiman Scholarship, NUI Galway, AO Foundation and Science Foundation Ireland for funding the research of Zepur Kazezian. Additionally, Authors also acknowledge College of Medicine Nursing and Health Sciences Scholarship, NUI Galway and the European Regional Development Fund and Science Foundation Ireland under Ireland's European Structural and Investment Fund, Grant Number 13/RC/2073, for supporting the Ph.D. studies of Kieran Joyce. All figures were created using biorender.com.

**Conflicts of Interest:** Their authors have no conflict of interest.

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
