# Peer review of "The Role of Hyaluronic Acid in Intervertebral Disc Regeneration"

_applsci, doi:10.3390/app10186257_

Round 1

Reviewer 1 Report

  1. Please strengthen the discussion of LBP therapeutic methods in clinical and their effaciacy.
  2. Please compare all LBP therapeutic methods in clinical by SWOT assay.
  3. Line 81: please present the full name of DRGs.
  4. LMW: please describe the molecule weight in detail.
  5. Table 1: Please correct NF-kb
  6. Please increase the resolution of all figures.

Author Response

Dear Editor and Reviewers: Thank you for reviewing our manuscript and providing us with valuable comments. We have revised the article to address all reviewers’ remarks, and the suggested alterations are incorporated in the revised manuscript. Our significant revisions include further discussion on low back pain (LBP) therapeutic strategies used in the clinic, additional discussion of current HA formulations and composite materials regarding the intrinsic properties of HA and minor revisions as per reviewer comments. A minor additional edit was the replacement of degenerative disc disease with IVD degeneration throughout the text.

REVIEWER 1:

  1. Please strengthen the discussion of LBP therapeutic methods in clinical and their efficacy.
  2. Please compare all LBP therapeutic methods in clinical by SWOT assay.

Response: We would like to thank the reviewer for their constructive feedback and comments. We amended the manuscript accordingly by adding the below paragraph on P.2 line 52 and have highlighted the respective sections. The authors are of the opinion that conducting a SWOT analysis of current IVD therapeutic strategies is complex. The heterogenous presentation and management of patients make such a comparison difficult. Furthermore, due to the lack of consensus on patient stratification with the emerging use of HA as a treatment strategy, we are hesitant to speculate the forms of treatment HA formulations might replace. We have now provided a brief discussion on HA clinical trials and reported outcomes to expand on its clinical use.

“Treatment of LBP in the early and moderate stages is mostly monitored through conservative treatments including bed rest, physiotherapy and exercise [20, 21] which is followed by prescribing non-steroidal anti-inflammatory drugs (NSAID)s including Ibuprofen and COX-2 inhibitors (Celecoxib), that are effective in treating acute and chronic LBP comparing to placebo [22]. In addition to NSAIDs, muscle relaxants such as cyclobenzaprine are also used [23]. Moreover, Opioids and benzodiazepines are prescribed to treat LBP. However, they have a short-term effect and need to be prescribed for no longer than four weeks because of the risk of developing dependency [24, 25]. Furthermore, Epidural and systemic corticosteroids are recommended with clinical trials exhibiting heterogeneous results, with the latest outcomes indicating glucocorticoid receptor-specific steroids may be more effective comparing to mineralocorticoid-targeting steroids [26].

Although most of the LBP patients will recover after an acute phase; however, 10% may develop chronic low back pain [27]. In chronic stages, generally, surgery is required because the IVD is unable to cure itself [28]. Existing surgical interventions include discectomy and spinal fusion or, sometimes in the cervical spine, a total disc replacement [29]. Patients who had discectomy show improvement when compared to nonsurgical therapy in short-term follow-ups; however, long term follow-ups (beyond two years) indicate no significant difference in the outcomes [30]. Spinal fusion is the typical surgical approach for chronic LBP. Although nonsurgical treatment for chronic LBP may be useful in a particular cohort, lumbar fusion remains more effective in alleviating pain and decreasing disability than commonly used nonsurgical treatment [31]. Disc replacement is also used to tackle the incidence of the disease in the adjacent segments due to the acute variation in motion segment mechanics linked with anterior cervical discectomy and fusion [32]. There is minimal evidence that suggests early surgical intervention improves long-term results in patients with lumbar disc herniation and radiculopathy that did not progress yet into neurologic deficit [30, 33]”.

  1. Line 81: please present the full name of DRGs.

Response: We would like to thank the reviewer for this comment. We added the full name of DRGs (line 81).

  1. LMW: please describe the molecule weight in detail.

Response: We acknowledge the need for further explanation of the LMW fragments of HA which, unlike the HMW HA can lead to further inflammation of the disc. We amended the manuscript accordingly by:

The high molecular weight of HA (HMW) that is 2 – 4 x 106 Da is produced by HAS1 and HAS2, while the low molecular weight of HA (LMW) that is 1 × 105 - 1 × 106 Da is produced by HAS3 [63]. We also highlighted the new addition in the manuscript p3, lines 112-113.”

  1. Table 1: Please correct NF-κB.

Response: We would like to thank the reviewer for this suggestion. We corrected NF-κB in Table 1 and highlighted it.

  1. Please increase the resolution of all figures.

Response: We would like to thank the reviewer for the feedback. We updated the quality of the images by producing the figures in PNG format. We also worked further on figure 3 to match the two other figures and also to increase its quality. The updated figures in the PNG format are uploaded to the system.

Reviewer 2 Report

The manuscript by Kazezian described the role of hyaluronic acid (HA) in IVD regeneration. The signaling pathways associated with degeneration of IVD were described, followed by the effect of introducing high and low MW HA in IVD. Finally, the preclinical models for investigating IVD disease were introduced. Overall, the manuscript was well-written. My only comment is that Section 4 is too short. Instead of referring readers to study Table 2 which contains lots of information, the authors should summarize the main points in Section 4, such as the different biomaterials used to form composite materials with HA and highlights of one or two representative studies, the concept of cell delivery using HA-based carrier and significance of it, and example of drugs that can be delivered along with HA. The authors should also discuss the clinical trials involving HA to treat IVD degeneration in the manuscript, since they represent the forefront of HA research.

Author Response

We would like to thank reviewer 2 for his encouraging feedback and suggestion.

 We acknowledge the need for elaborating section 4 for helping the reader for a better read through the key points regarding the therapeutic effects of HA. We have revised the section 4 of the manuscript according to the suggestions and highlighted it in the original manuscript. Therefore, the below paragraph was added to section 4, p15.

There are an increasing requirement for three-dimensional (3D) scaffolds to regenerate IVDs. HA’s application in such 3D structures was diverse. Different materials were incorporated with HA to form composites with biocompatible properties including biological performance, stiffness, and degradation both in vitro and in vivo. Such composites included collagen, fibrin and CS [133, 134, 137, 147] that have been investigated in order to tackle the extracellular matrix disintegration by enhancing the synthesis of key matrix modulating proteins, GAGs and increasing cellular viability. Pre-clinical models of IVD degeneration to test the efficacy of HA-based materials have been developed in mice [146], rats [51], rabbit [134], sheep [144], goats [145], pigs [147], cows [133], and primates [135]. HA formulations have been cross-linked using many methods including UV cross-linking [133], divinyl sulfone [135], PEG-amine with EDC/NHS (carbodiimide) [50], fibrin [141], batroxobin [143], butanediol diglycidyl ether [148] and enzymatic crosslinking using transglutaminase in a fibrin/HA material [145]. HA hydrogels utilized in treating the different parts of the IVD including AF and NP not only acted as composite materials but also as a reservoir for growth factors and therapeutic drugs [133, 141, 145], as well as a cell delivery vehicle [138, 143, 146].

In summary, these studies report on the hydrophilic properties of HA to restore disc height [142] and MRI signal intensity [135], the ability of HA to promote ECM synthesis [138] in resident or injected cells, and of course the anti-inflammatory [136] and anti-nociceptive [139] properties of HA in the setting of IVD degeneration. Several clinical trials investigating the efficacy of HA-based materials are ongoing, with few reporting results at this early stage. However, initial results have been promising, where MSCs delivered in a HA carrier system has demonstrated an excellent safety profile with several patients reporting improved VAS and ODI reflecting better MRI signal intensity [148]. Furthermore, HA is being trialled in extra-discal injections in cases of facet joint degeneration [149]. HA has demonstrated equivalent efficacy to glucocorticoid administration in this patient cohort. Many studies are yet to report results, including a bone marrow derived-MSC (BM-MSC)/HA system in a Phase 2 trial (NCT01290367), Rexlemestrocel-L combined with hyaluronic acid in Phase 3 (NCT02412735) and discogenic cells in a HA delivery system with study arms in USA and Japan in Phase 1 (NCT03347708). There is much excitement surrounding these systems, given the proven HA pre-clinical efficacy.